# DERA in Flow: Synthesis of a Statin Side Chain Precursor in Continuous Flow Employing Deoxyribose-5-Phosphate Aldolase Immobilized in Alginate-Luffa Matrix

**Bianca Grabner \*, Yekaterina Pokhilchuk and Heidrun Gruber-Woelfler \*** 

Institute of Process and Particle Engineering, Graz University of Technology, Inffeldgasse 13/III, 8010 Graz, Austria; y.pokhilchuk@student.tugraz.at
\* Correspondence: b.grabner@tugraz.at (B.G.); woelfler@tugraz.at (H.G.-W.)

**Abstract:** Statins, cholesterol-lowering drugs used for the treatment of coronary artery disease (CAD), are among the top 10 prescribed drugs worldwide. However, the synthesis of their characteristic side chain containing two chiral hydroxyl groups can be challenging. The application of deoxyribose-5-phosphate aldolase (DERA) is currently one of the most promising routes for the synthesis of this side chain. Herein, we describe the development of a continuous flow process for the biosynthesis of a side chain precursor. Design of experiments (DoE) was used to optimize the reaction conditions (pH value and temperature) in batch. A pH of 7.5 and a temperature of 32.5 °C were identified to be the optimal process settings within the reaction space considered. Additionally, an immobilization method was developed using the alginate-luffa matrix (ALM), which is a fast, simple, and inexpensive method for enzyme immobilization. Furthermore, it is non-toxic, biodegradable, and from renewable resources. The final continuous process was operated stable for 4 h and can produce up to 4.5 g of product per day.

**Keywords:** immobilized DERA; statin side chain; continuous flow synthesis; alginate-luffa matrix; design of experiments; optimization

## 1. Introduction

Cardiovascular diseases (CVD) are the number one cause of death worldwide [1]. In Europe each year, 3.9 million deaths (45% of all deaths) are associated with CVD [2]. The primary reason for death among CVD patients is coronary artery disease (CAD). CAD is characterized by arthrosclerosis—the formation of sedimentation (plaque) in the blood vessel that leads to reduced oxygen supply to the heart [3]. Arterial plaque formation is often attributed to an increased level of low-density lipoprotein (LDL) cholesterol, which in Western culture is often caused by unhealthy foods and little exercise. Beside a change in lifestyle being the first choice for lowering the cholesterol level in blood, one can choose between three medical mechanisms for the treatment of hypercholesterolemia increase bile synthesis, decreased intestinal cholesterol absorption, or inhibition of the 3-hydroxy-3-methylglutary coenzyme A (HMG-CoA) reductase, an essential enzyme in the synthetic route to cholesterol [4,5]. Statins competitively inhibit HMG-CoA reductase and thus approach the reduction of LDL cholesterol concentration by the latter of the above-mentioned methods. One of the most prevalent statins is atorvastatin (Lipitor®), synthesized by Pfizer since 1996. It is known as the best-selling blockbuster in the past two decades [6]. Despite the patent expiration in 2011, Lipitor® was the third most commonly prescribed medication in the U.S. in 2016. An additional statin, simvastatin, was the eighth most prescribed medication; therefore, statins represent a significant share of the pharmaceutical market [6].

Natural and semi-synthetic statins possess side chains in the form of lactones, which are in vivo hydrolyzed to the corresponding and biologically active hydroxyl acid. Synthetic statins, so-called super-statins such as rosuvastatin (Crestor®), are provided in the active form of dihydroxy heptanoic acid with two chiral alcohol groups attached to a heterocyclic core [7,8]. The structures of the three above-mentioned statins are depicted in Figure 1.

**Figure 1.** Molecular structures of three statins: Simvastatin, atorvastatin, and rosuvastatin. Dihydroxy heptanoic acid side chain and its cyclic precursor are colored in red.

Only one of the enantiomers of the chiral side chain is active and needs to be provided in high purity for adequate activity. This is a major challenge for manufacturers [9,10]. In the past decade, numerous approaches for the enantiomerically pure synthesis of this side chain were published [11–15]. Chemical routes requiring harsh chemicals and numerous additional steps for the protection/de-protection of sensitive functional groups, by-product formation, and waste generation are an issue. In contrast to this, it was shown by Tao et al. that biocatalysis could be a comparably sustainable approach [16,17]. Beside numerous chemo-enzymatic routes, the one employing deoxyribose-5-phosphate aldolase (DERA, EC 4.1.2.4) is very promising. DERA is a unique enzyme able to catalyze the aldol addition of two aldehydes resulting in an aldehyde product, which can again serve as a substrate for another aldol addition. The product after two sequential addition reactions and spontaneous cyclization is the hemiacetal 2,4,6-trideoxyhexose **1c** (Figure 2). This unique property of DERA was discovered by Gijsen et al. in 1994 and extensively studied around the millennium [18–23]. The product of this biotransformation can be further processed to the statin side chain via the oxidation and subsequent ring-opening of **1e** (Figure 2). The application of DERA on an industrial scale is challenging as the enzyme is sensitive to high concentrations of acetaldehyde, its natural substrate. The active site of the wildtype DERA (DERA$_{WT}$) is irreversibly inhibited by the covalent binding of the side-product, crotonaldehyde. In 2016, the group of Pietruszka at the Research Center Jülich GmbH tackled this issue and developed a mutant (C47M), which is resistant to acetaldehyde to a high degree [24]. This mutant showed outstanding catalytic activity in tests using acetaldehyde as the donor molecule. Further implementation of the mutant was not reported by the group.

In the present work, a continuous process for the synthesis of a statin side chain precursor was developed. Continuous processes go hand in hand with a number of advantages such as reduced reaction time, constant product quality and cost reduction, compared to batch mode. While continuous operations are well implemented in bulk industries such as paper and food, they have barely made their way into pharmaceutical drug synthesis [25]. Fortunately, numerous researchers are passionate about continuous flow synthesis and aim to accelerate the establishment of continuous processes in the industry [26–29]. In this manuscript, we describe the process of developing a continuous biocatalytic synthesis employing the novel DERA mutant. Immobilized freeze-dried whole cells (*Escherichia coli* hosting DERA (C47M)) in an alginate gel matrix on a luffa sponge were used in a packed-bed reactor. This immobilization method was originally developed by Phisalaphong et al. and named alginate-loofa matrix (ALM) [30]. This method is simple, inexpensive, and fast in preparation. Loofa sponge (*Luffa cylindrical*) as support brings a number of advantages. The matrix originates from a renewable source, is a highly porous material, and is fully biodegradable [31–33].

**Figure 2.** Deoxyribose-5-phosphate aldolase (DERA)-catalyzed stereoselective aldol addition of three aldehydes to produce a lactol (**1c**), which can be further oxidized to a lactone results in the typical statin side chain after ring-opening.

In this work, design of experiments (DoE) was applied for the optimization of pH value and temperature for the enzymatic reaction and optimizing the flow conditions. DoE is a multivariate approach for parameter screening and optimization. In contrast to the original approach, where one factor is changed at a time, this method allows the identification of the interaction between the individual parameters. DoE helps in gaining maximum information from a minimum number of experiments [34–36].

So far, no such development process using DoE for the optimization of a continuous enzymatic process was described in the literature, to the best of our knowledge. Furthermore, this immobilization method and the application of DERA in continuous flow processes are barely found in the literature.

## 2. Results and Discussion

### 2.1. Design of Experiments for Optimal Batch Conditions

In order to operate the continuous process under ideal conditions, the optimal parameters for the enzymatic reaction needed to be determined. For that, design of experiments (DoE) was used to evaluate the effect of the two crucial process parameters, temperature, and pH value on intermediate and product formation. In the first circuit of experiments, a rough, full-factorial lattice was designed. The temperature ranged between 28 °C and 37 °C in steps of 4.5 °C and the pH ranged between 6.0 and 8.0 in steps of 0.5 in order to screen a wide range of process settings. For the second circuit, a fine full-factorial lattice was laid in the optimum of the response surface of the first experimental circuit. Both designs are shown in Figure 3.

For each point on the lattice, an experiment was conducted. On a 500 μL scale, 1.5 M of acetaldehyde was dissolved in 0.1 M of TEOA buffer set to the respective pH value via HCl. Samples were collected over time and analyzed by means of GC-FID. The collected data (reaction rate for intermediate and product formation and enzyme stability) was evaluated using MATLAB®. Details on the experimental design and parameters can be found in ESI (Electronic Supplementary Information).

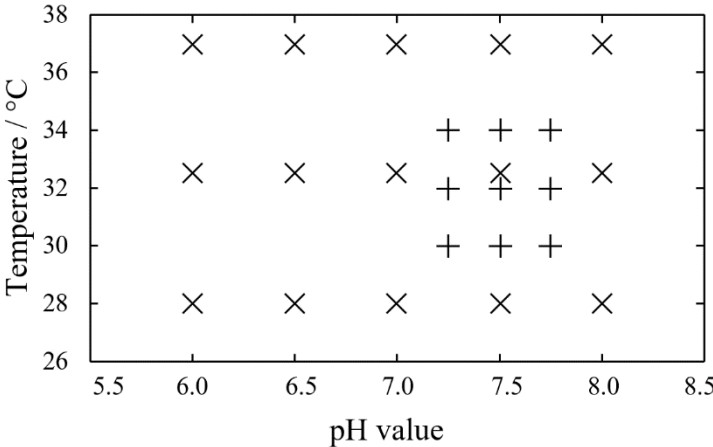

**Figure 3.** Experimental design chosen for the two circuit of design of experiments (DoE) (1st circuit: x; 2nd circuit: +).

The response surface for the rate for product formation shown in Figures 4 and 5 was obtained from the contour plot in Figure S9. It shows an almost linear incline in enzyme activity with increasing reaction temperature. This correlation between the temperature and reaction rate is also clearly visible in Figure S7 in ESI, which shows the data points without surface. The shape of this surface can be described by the Arrhenius law and the rate of inactivation, which is mathematically a logarithmic function. The productivity of the catalytic system grows until it reaches the point where the inactivation rate is higher than the reaction rate [37]. The highest activity is supposed to be at 34.5 °C. However, the deactivation rate due to denaturation is also high at this temperature. At 34 °C, the activity was reduced by 30% after 1 h of reaction time, while a reaction temperature of 32.5 °C retained more than 95% of the initial activity after 1 h reaction time. Therefore, for all the following experiments 32.5 °C, was the temperature of choice, as it constitutes the ideal compromise between the required reaction rate for a continuous application and stability for a steady state over several hours. The connection between pH value and reaction rate shows the typical Gaussian-like distribution, with an optimum value at pH 7.5 (Figure 5).

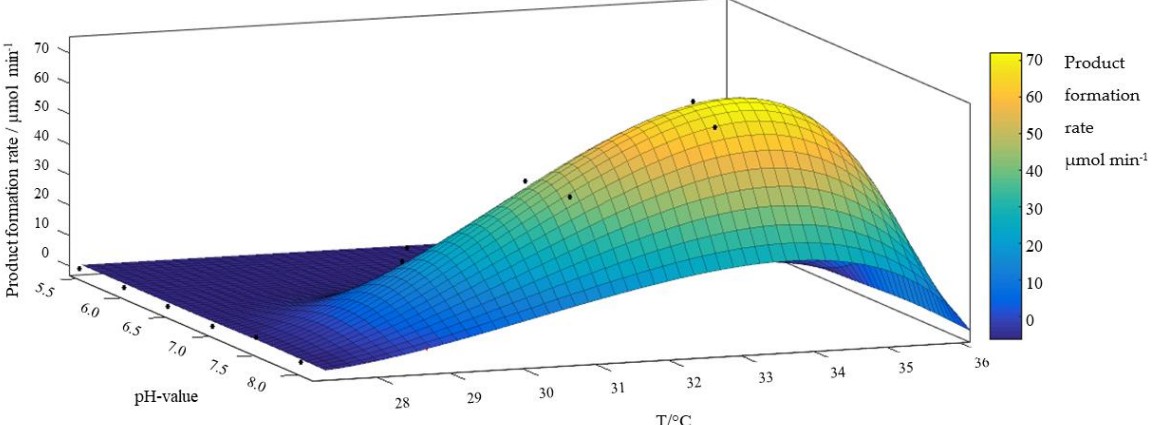

**Figure 4.** Response surface (view from the perspective over T) for the experimental design over T. An amount of 1.5 M of **1** in a final volume of 500 μL, 0.1 M of buffer (6.0 ≤ pH ≤ 8.5), 7 μL of DMSO, and 10 mg of freeze-dried *E. coli* cells hosting DERA, 28 °C ≤ T ≤ 37.

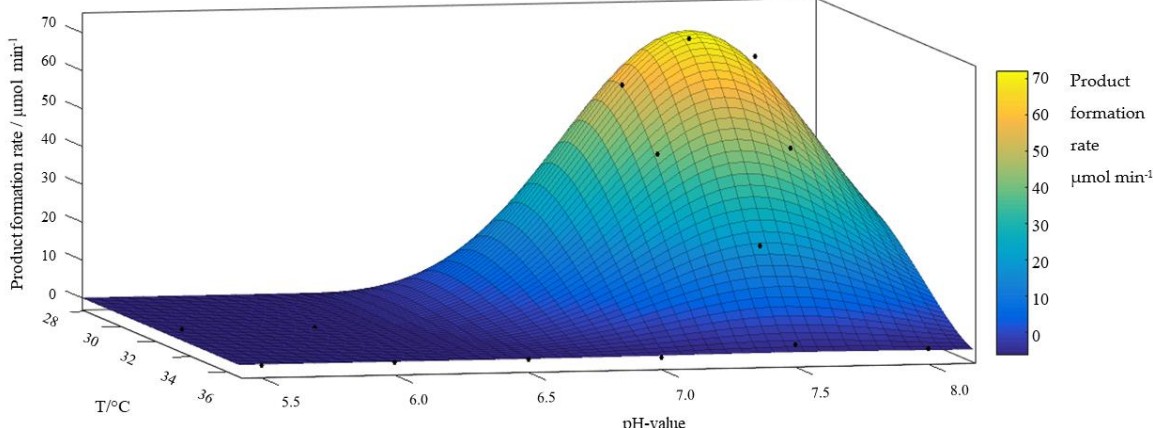

**Figure 5.** Response surface (view from the perspective over pH) for the experimental design over pH value. An amount of 1.5 M of **1** in a final volume of 500 μL, 0.1 M of buffer (6.0 ≤ pH ≤ 8.5), 7 μL of DMSO, and 10 mg of freeze-dried *E. coli* cells hosting DERA, 28 °C ≤ T ≤ 37.

At pH ≤ 7.0, no lactol was formed at all. Based on the results of the DoE, all the following experiments were conducted at 32.5 °C and pH 7.5.

## 2.2. Substrate Screening

Five substrates were tested under optimized batch conditions for their potential to serve as an acceptor for aldol addition (Figure 6). Acetaldehyde **1** and its chloro derivative **2** were converted to the desired products, while larger residues (benzaldehyde **4** and cinnamaldehyde **5**) and acrolein **3** were not accepted. The major issue with aromatic substrates is the limited solubility in aqu. buffer systems, even in mixtures with DMSO [38]. These substrates would be especially interesting, as the direct addition of acetaldehyde to the molecule core would reduce the number of process steps drastically. Acrolein was also tested as substrate. Although the mutant was successfully tested for its tolerance toward crotonaldehyde, acrolein still inhibits the enzyme [38,39]. Further investigations were conducted using acetaldehyde **1** and chloroacetaldehyde **2**.

**Figure 6.** Overview of tested substrates and corresponding products.

## 2.3. Kinetics

After the optimal reaction conditions and accepted substrates were identified, it was of interest to investigate the kinetic behavior of the addition reaction. This reveals important information required for designing the continuous process. First, a batch experiment in which three molecules of acetaldehyde **1** are linked to the lactol **1c** was conducted. It shows that product formation is initiated as soon as intermediate concentration exceeds 100 mM (Figure 7). After 3 h, the conversion exceeded 95% and the yield (determined by GC-FID) reached 88.5%.

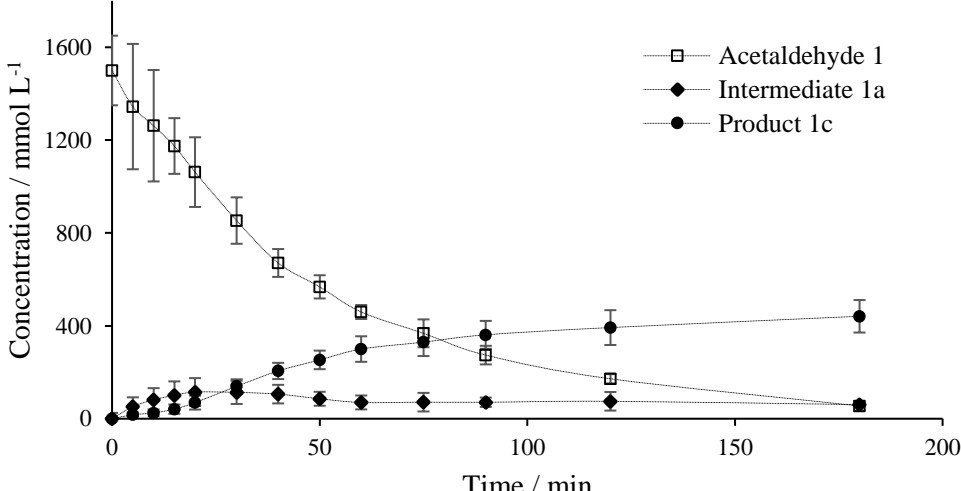

**Figure 7.** Time course of conversion of acetaldehyde **1** with DERA to the dimer **1a**, and subsequently to the lactol **1c**, at the optimized reaction conditions (32 °C, pH 7.5) based on the DoE, 5 mL of final volume, 0.1 M of TEOA buffer, 1.5 M of substrate concentration, 70 μL of DMSO, and 100 mg of DERA in freeze-dried *E. coli* cells.

As **1** and thus **1c** do not host any functional groups that could serve in further coupling reactions to link the product to the core of the API, the chloro derivative **2** is of greater interest for the industry. Fortunately, **2** proved to be converted faster than acetaldehyde. Most of the reaction progress can be observed within the first 60 min of the reaction, after which a 75% yield was detected (Figure 8). After 3 h, more than 90% was reached. For the continuous flow application, **2** was chosen as the aldehyde acceptor for the reasons mentioned above.

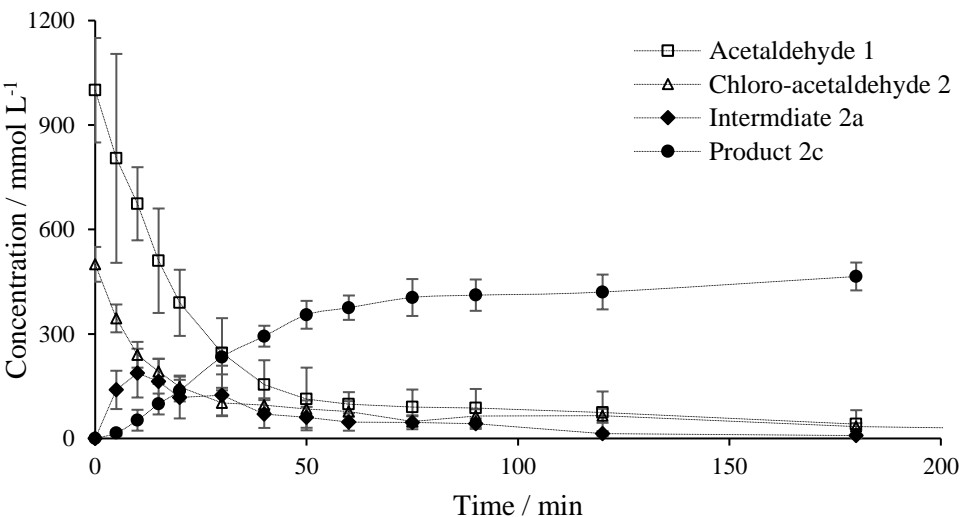

**Figure 8.** Time course of batch bioconversion of chloroacetaldehyde 2 (0.5 M) and acetaldehyde 1 (1.0 M) to the dimer 2a, and subsequently to the lactol 2c, at the optimized reaction conditions (32 °C, pH 7.5) based on the DoE, 5 mL of final volume, 0.1 M of TEOA buffer, 70 μL of DMSO, and 100 mg of DERA in freeze-dried *E. coli* cells.

### 2.4. Immobilization of DERA in Alginate-Luffa Matrix

Since the use of a packed-bed reactor brings along a number of advantages over the application of homogeneous catalysts, an appropriate immobilization method for the enzyme was required. Covalent binding to a solid support needs an additional purification step of the enzyme prior to linking, which is a labor-intense process. In order to keep catalyst preparation simple, adsorption

and encapsulation were the remaining options. Adsorption bears the risk of enzyme leaching due to loose binding. Encapsulation into a matrix is a fast, inexpensive, and simple technique to immobilize isolated biocatalysts or whole cells.

Alginate is non-toxic, biodegradable, and made from a renewable feedstock (bacteria), and thus fulfills all requirements for a "green" matrix for biocatalyst encapsulation. Freeze-dried *E. coli* cells hosting overexpressed DERA were immobilized in two ways in alginate beads and an alginate of luffa sponge. The former of the two was prepared by dissolving 2% (*w/v*) Na-alginate in 2 mL of a 0.9% (*w/v*) NaCl solution. After a clear solution was obtained, the biocatalyst was suspended in the viscose liquid. To form spherical beads from the suspension, the mixture was dropwise added to a 2% (*w/v*) solution of a bivalent cation for cross-linking, $Ca^{2+}$ ($CaCl_2$) or $Ba^{2+}$ ($BaCl_2$). Other tested cations ($Zn^{2+}$, $Fe^{2+}$, and $Mg^{2+}$) did not lead to sufficient cross-linking to form a stable alginate matrix. Since the alginate matrix displays a barrier for the substrate, which results in a reduction of reaction rate, an increase in surface can be beneficial for the reaction rate. For that, a porous material serving as support, which could be coated with the alginate matrix enclosing the enzyme, was desired. A luffa sponge was chosen as support because it is a natural product of high porosity. This immobilization technique is called the alginate-luffa matrix (ALM). In order to immobilize the same volume of an alginate-enzyme mixture on the luffa sponge, a volume of 2.5 $cm^3$ (245 mg) was required. The sponge was soaked with the cell suspension and cross-linking was induced either by $Ba^{2+}$ or $Ca^{2+}$ (details in ESI). The results of the ALM were compared with cells immobilized in conventional alginate beads (I.D. 2 mm) (Figure 9). The comparison shows that ALM is four times more active than the beads. The type of cross-linking cation has hardly any effect on the performance of the enzyme. ALM enclosing DERA was further employed for application in continuous flow.

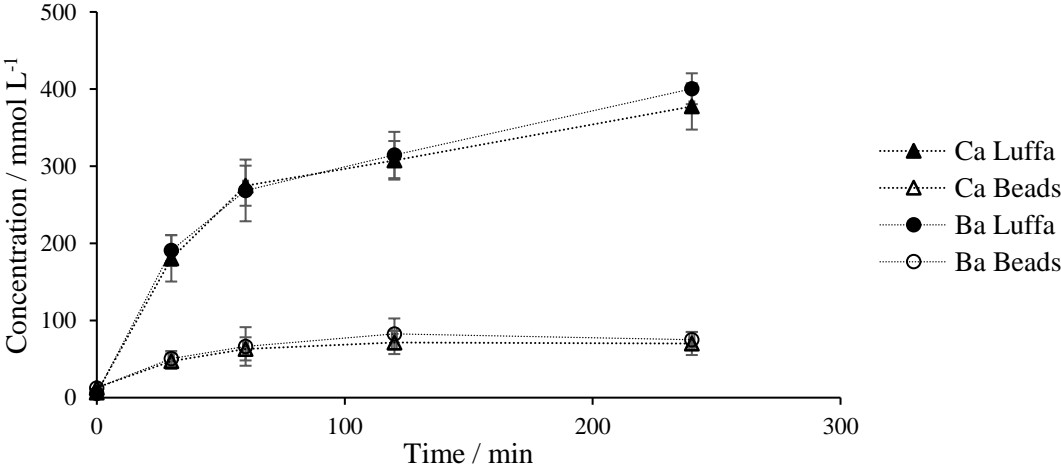

**Figure 9.** Time course of lactol formation by immobilized DERA on Luffa or in beads, 5 mL of final volume, 0.1 M of TEOA buffer with pH 7.5, 70 mL of DMSO, 0.5 M of **2**, 1.0 M of **1**, 32.5 °C, 100 mg of freeze-dried *E. coli* cells hosting DERA, 2% (*w/v*) Na-alginate in 0.9% (*w/v*) NaCl solution, cross-linking induced by $Ca^{2+}$ or $Ba^{2+}$, 2% (*w/v*), respectively.

## 2.5. Flow Application

Flow experiments were carried out in the so-called "Plug and Play" reactor [40]. It is a modular bench-top reactor (20 cm × 15 cm) equipped with heating/cooling shell and openings for reaction modules (commercially available HPLC columns). For catalyst immobilization, a cylindrical piece of the sponge fitting into the column was cut manually to serve as solid support. Eight hundred and fifty milligrams (10 $cm^3$) of luffa was fitted into the column. Immobilized DERA on the loofa sponge was packed in an HPLC column (20 cm × 8 mm) (Figure 10), which was placed in the reactor. After heating the catalyst to the respective temperature, the bed was flushed with buffer in order to remove residual chemicals from the immobilization process prior to running the reaction.

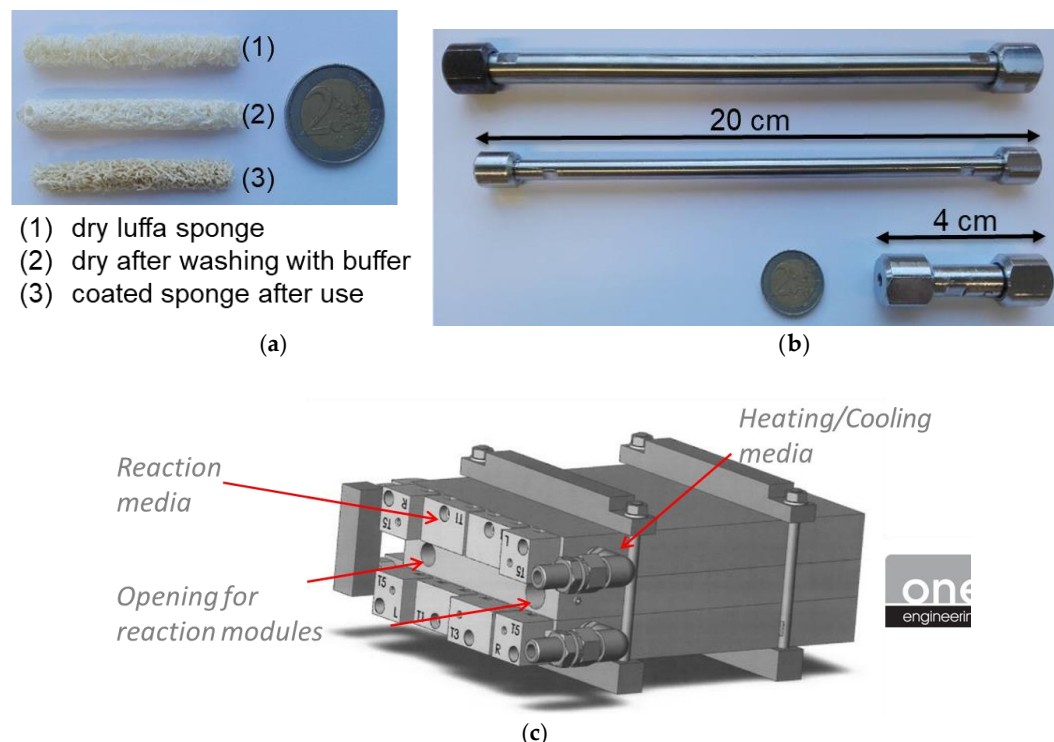

**Figure 10.** Equipment used for continuous synthesis: luffa sponge (**a**), stainless steel tube (**b**), and "Plug and Play" reactor (**c**).

### 2.5.1. Residence Time Distribution

Prior to conducting experiments in flow, the mean residence time in the reactor for various flow rates was determined. For that, the column was packed with a luffa sponge and the mean residence time was determined (details in ESI). The results are summed up in Table 1.

**Table 1.** Theoretically and experimentally determined mean residence times and bodenstein (Bo) numbers for the three test flow velocities going through an HPLC column (20 cm × 0.8 cm) packed with a loofa sponge.

| $v$ | $\bar{t}_{th}$ [min] | $\bar{t}_{exp}$ [min] | Bo [–] |
|------|------|------|------|
| 0.10 | 60.5 | 62.7 | 15.0 |
| 0.25 | 24.2 | 26.8 | 9.5 |
| 0.50 | 12.1 | 11.4 | 33.3 |

$\bar{t}_{th}$ = theoretical calculated mean residence time, $\bar{t}_{exp}$ = experimentally determined mean residence time. See Electronic Supplementary Information (ESI) for details.

### 2.5.2. Design of Experiments for Continuous Flow Application

For optimizing the flow process, the flow rate and substrate concentration were of interest. Furthermore, a potential effect of the cation used to induce cross-linking of alginate on the catalyst performance should be investigated. A full-factorial design for three parameters (flow rate, concentration, cation) was developed. Three levels were set for flow rate (0.1, 0.25, and 0.5 mL/min) and substrate concentration (0.25, 0.5, and 0.75 M of **2a**). The influence of the chosen cross-coupling ion was investigated on two levels ($Ba^{2+}$ and $Ca^{2+}$). The performance of the process was followed by collecting samples at the outlet of the reactor and determining the yield by means of GC-FID. The result was evaluated using MODDE® (Figures 11 and 12).

DERA immobilized in alginate cross-linked with Ba$^{2+}$ showed stronger dependence on the flow rate and also on substrate concentration. At high substrate concentration, this catalyst is less active. High flow rate leads to low yield. Details on the results of the DoE can be found in ESI.

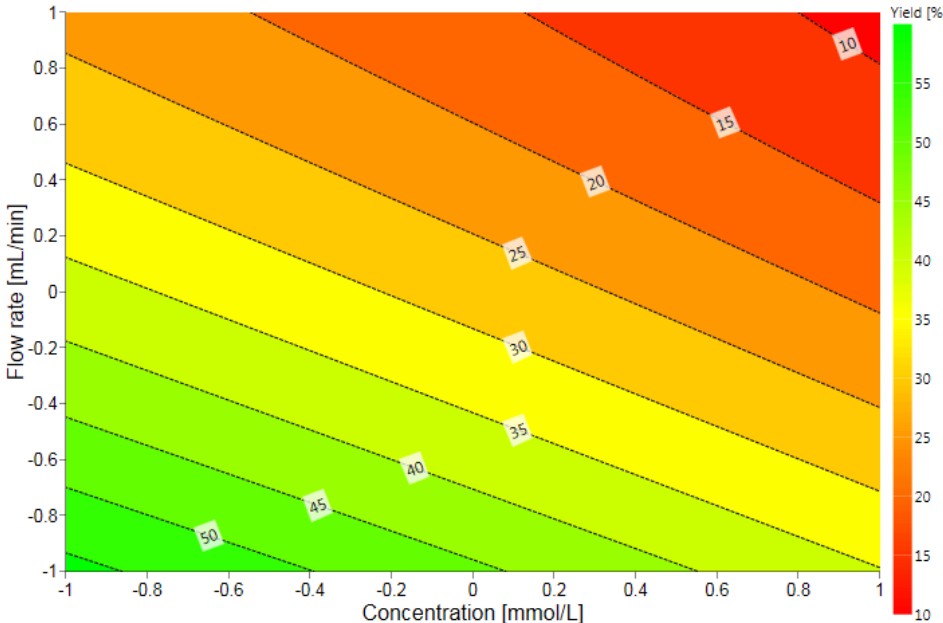

**Figure 11.** Contour plot for the performance of the continuous flow process for Ba$^{2+}$ cross-linked alginate. Stock: 250–750 mM of **2**, 500–1500 mM (2 mol-eq.) of **1** in 0.1 M of TEOA buffer, pH 7.5, with 1.4% of DMSO. T = 32.5 °C; flow rate: 0.1–0.5 mL/min; 350 mg of freeze-dried *E. coli* hosting DERA on 850 mg of luffa sponge (10 mL of 2% (*w/v*) Na-alginate solution); cross-linking by 2% (*w/v*) BaCl$_2$ solution.

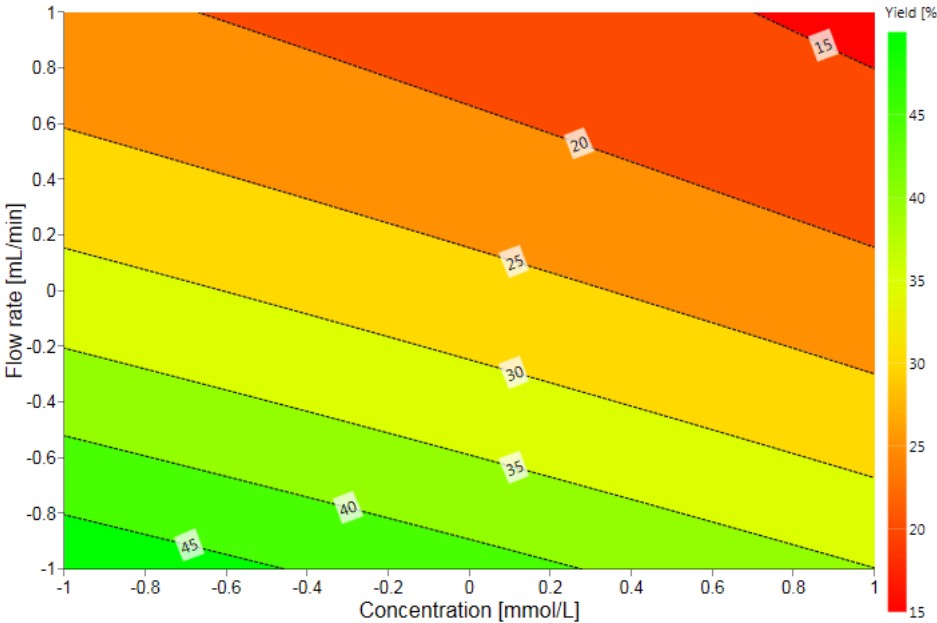

**Figure 12.** Contour plot for the performance of the continuous flow process for Ca$^{2+}$ cross-linked alginate. Stock: 250–750 mM of 2; 500–1500 mM (2 mol-eq.) of **1** in 0.1 M of TEOA buffer, pH 7.5, with 1.4% of DMSO. T = 32.5 °C; flow rate: 0.1–0.5 mL/min; 350 mg of freeze-dried *E. coli* hosting DERA on 850 mg of luffa sponge (10 mL of 2% (*w/v*) Na-alginate solution); cross-linking by 2% (*w/v*) CaCl$_2$ solution.

Ca$^{2+}$ cross-linked alginate was found to not immobilize the catalyst sufficiently, which led to enzyme leaching and rapid loss of catalyst activity within the reactor. Enzyme leaching was proved by not quenching the samples collected at the outlet of the reactor. The reaction proceeded, indicating the presence of active enzymes in the reaction mixture leaving the column. Ba$^{2+}$ cross-linked alginate did not suffer from enzyme leaching. Therefore, this catalyst could also be applied for an increased period of time in a continuous process.

### 2.5.3. Continuous Synthesis

As a final step, the ideal reaction conditions obtained from the DoE approach were applied for a continuous flow process for the synthesis of a statin side chain precursor. The time course for the process output is shown in Figure 13. An increase in enzyme loading from 350 mg to 700 mg in the reactor led to almost an 80% yield of **2c**. However, higher cell concentration in the alginate solution resulted in a highly viscous mixture, which made the handling and immobilization process more difficult.

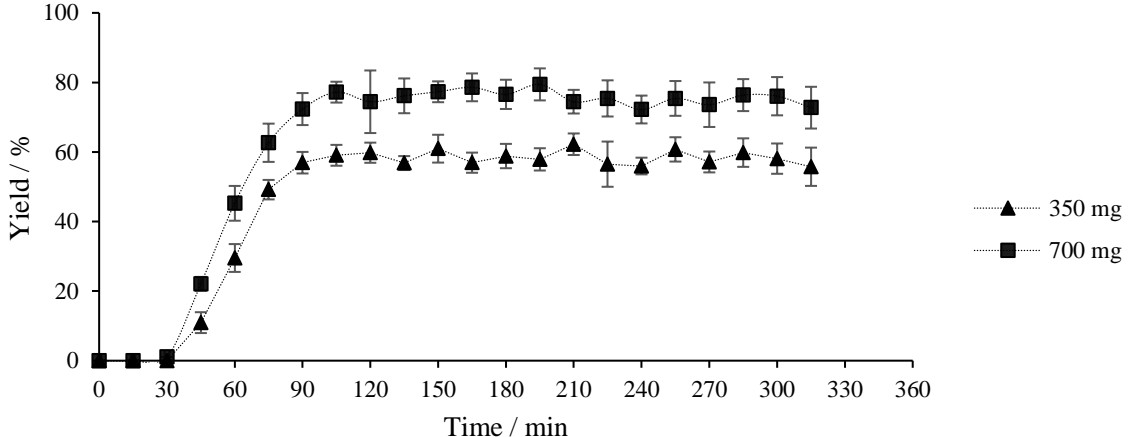

**Figure 13.** Yield over time for continuous flow process using 350 mg and 700 mg of freeze-dried cells hosting DERA; 32.5 °C, 0.1 M of TEOA buffer, pH 7.5, 0.1 mL/min flow rate, 0.25 M of **2**, 0.5 M of **1**.

## 3. Materials and Methods

### 3.1. General

All chemicals used were purchased from Sigma Aldrich in analytical grade and used as received.

### 3.2. Enzyme Expression and Purification

Two microliters of plasmid isolation and 100 µL of competent *Escherichia coli* cells (strain BL21(DE3)) were combined in a 0.2 cm electroporation cuvette and placed in the BioRad MicroPulser set to "Ec2". The mixture was pulsed once. One milliliter of SOC medium (0.5% (*w/v*) yeast extract, 2% (*w/v*) tryptone, 10 mM of NaCl, 2.5 mM of KCl, 20 mM of MgSO$_4$, pH 7.5) was added and the cells were suspended in the medium. After incubating for 1 h at 37 °C, the cells were applied on agar plates containing the antibiotic Ampicillin and were grown overnight at 37 °C. On the following day, one culture spot was transferred into a shaking flask together with 200 mL of LB medium (5 g/L of yeast extract, 10 g/L of tryptone, 10 g/L of NaCl) and 200 µL of Ampicillin. The culture was incubated overnight at 37 °C. The cells were harvested by centrifugation (15 min, 7000× *g*, 4 °C) and the obtained pellets were freeze-dried.

### 3.3. Optimization of Reaction Conditions in Batch Using Design of Experiments (DoE)

In the first round, a rough, full-factorial lattice with two variables (temperature and pH) was designed. The temperature ranged between 28 and 36 °C in steps of 4.5 °C, while the pH was set to a

value between 6.0 and 8.0 in steps of 0.5. Each reaction was carried out with 10 mg of freeze-dried cells hosting DERA suspended in 500 μL of buffer with 1.5 M of acetaldehyde as substrate and 7 μL of DMSO [24]. The mixture was stirred with 200 rpm at the respective temperature. Product formation was followed by taking samples after 30 and 60 min. For the samples, 100 μL of reaction suspension were mixed with 400 μL of acetonitrile to stop the reaction by precipitating the enzyme. After separating the inactivated enzyme by centrifugation (10 min, 15,000 rpm), the clear solution was transferred into a GC vial and analyzed by means of GC-FID. The results (yield of intermediate and product) were evaluated by surface fitting in MATLAB® using the least square method. A Gaussian function was the result for the description of the activity over pH, while a combination of a polynomial function (increasing activity with rising temperature) and a logarithmic function (describing inactivation at elevated temperature) fitted the surface over temperature. From the response surface, a new, narrow, and full-factorial lattice was planned. This time, the temperature ranged between 30 and 34 °C (steps of 2 °C) and the pH was adjusted to 7.25, 7.5, or 7.75. The reaction and analysis were carried out the same way as for the first round.

### 3.4. GC Analysis

Samples were analyzed by means of gas chromatography (GC) using a Perkin Elmer (USA) Clarus 500 equipped with an Optima 5-MS 0.25 μm, 30 m × 0.32 mm ID capillary column, and a flame ionization detector (FID) run on $H_2$ and synthetic air. $N_2$ was used as carrier gas. The heating program was set as follows: initial temperature of 50 °C (5 min), and gradient of 10 °C min$^{-1}$ to 250 °C (5 min). Injection volume: 1 μL. This method was adapted from the literature [41]. Retention times: **1**, 1.6 min; **1a**, 3.3 min; **1b**, 10.2 min; **1c**, 11.9 min; **2**, 2.0 min; **2a**, 7.9 min; **2c**, 15.4 min.

### 3.5. Substrate Screening

One hundred milligrams of freeze-dried cells were suspended in 5 mL of 0.1 M of triethanol amine (TEOA) buffer, pH 7.5. After heating to 32.5 °C, 70 μL of DMSO, the substrates (500 mM of acceptor 1-5, respectively, and 1 M of acetaldehyde 1) were added. The reaction mixture was stirred with 600 rpm. Samples were taken over time. For each sample, 200 μL of the reaction mixture were quenched with 800 μL of acetonitrile. Centrifugation (10 min, 15,000 rpm) led to good separation of the precipitated catalyst. The supernatant was transferred into a GC vial and analyzed by means of GC-FID.

### 3.6. Immobilization in Beads

A 2% *w/v* sodium alginate solution was formulated by dissolving 40 mg of Na-alginate in 2 mL of 0.9% *w/v* aqueous NaCl solution. One hundred milligrams of lyophilized *E. coli* cells were added and the mixture was stirred to homogeneity. In order to form beads, the mixture was then added dropwise to a 2% (*w/v*) solution of the cross-linking cation ($CaCl_2$, $BaCl_2$, $ZnCl_2$, $MgCl_2$, or $FeCl_2$) using a syringe and a needle. The beads were stirred in the cation solution for 60 min to let them solidify. The size of the beads could be varied by the diameter of the needle on the syringe. After filtering, the beads were washed with a 0.9% (*w/v*) NaCl solution and kept under ambient conditions for 30 min to let the surface solidify and become more resistant to mechanical abrasion.

### 3.7. Immobilization in ALM for Batch Reactions

A 2% *w/v* sodium alginate solution was formulated by dissolving 40 mg of Na-alginate in 2 mL of 0.9% *w/v* aqueous NaCl solution. One hundred milligrams of lyophilized *E. coli* cells were added and the mixture was stirred to homogeneity. Afterward, the carrier (luffa sponge, 245 mg cut into pieces 2.5 cm$^3$ in volume) was soaked in the mixture and transferred to a 2% (*w/v*) cation solution ($CaCl_2$, $BaCl_2$) where it was gently stirred for 60 min to solidify. The loaded carrier was then washed with 0.9% NaCl solution, kept at ambient conditions for 30 min to solidify, and stored in purified water at 4 °C until usage.

### 3.8. Immobilization in ALM for Flow Application

The loofa sponge was cut to completely be packed into the 20 cm × 0.8 cm HPLC column. The immobilization process is illustrated in Figure S11 in ESI. Three hundred and fifty milligrams or 700 mg of freeze-dried *E. coli* cells were suspended in a 2% (*w/v*) (200 mg) Na-alginate in 10 mL of 0.9% (*w/v*) NaCl solution (Figure S10a). Eight hundred and fifty milligrams (10 cm$^3$) of cylindrically cut luffa sponge was used to soak up the mixture. After completely soaking up the entire DERA-alginate solution, the loofa sponge was submerged into 2% *w/v* CaCl$_2$ or BaCl$_2$ for 1 h with stirring at room temperature for cross-linking, followed by 30 min of air-drying (Figure S10b). The resultant loofa sponge carrying immobilized freeze-dried whole cells entrapped by alginate was used to pack the HPLC column for further use in continuous flow experiments.

### 3.9. Design of Experiments for Optimizing Flow Process

Continuous experiments were carried out in the so-called "Plug and Play" reactor [40]. The 850 mg loofa sponge with the immobilized enzyme (350 mg of cells) was packed in the stainless steel column (20 cm × 8 mm). The reaction medium consisting of 0.25, 0.5, or 0.75 M of chloroacetaldehyde **2** and 2 mol-eq. of acetaldehyde **1**, with respect to **2**, in 0.1 M of TEOA buffer, pH 7.5, was pumped through the column using an HPLC pump (Knauer, Azura P4.1 S) set to 0.10, 0.25, and 0.50 mL/min, respectively. The reaction temperature was 32.5 °C. A sample was taken every 15 min by collecting an aliquot of 200 μL of the product stream and diluting it with 800 μL of acetonitrile. After centrifugation (10 min, 15,000 rpm), the supernatant was transferred to a GC vial and analyzed by means of GC-FID. The results were evaluated by means of MODDE$^{®}$. Details (table of experiments and results) are available in ESI.

### 3.10. Continuous Synthesis

The 850 mg loofa sponge with the immobilized enzyme (350 mg of cells) was packed in the stainless steel column (20 cm × 8 mm). The reaction medium, consisting of 0.25 M of **2** and 0.5 M of **1** in 0.1 M of TEOA buffer pH, 7.5, was pumped through the column using an HPLC pump (Knauer, Azura P4.1 S) set to 0.10 mL/min. The reaction temperature was 32.5 °C. A sample was taken every 15 min by collecting an aliquot of 200 μL of the product stream and diluting it with 800 μL of acetonitrile. After centrifugation (10 min, 15,000 rpm), the supernatant was transferred to a GC vial and analyzed by means of GC-FID.

## 4. Conclusions

We were able to develop an optimized continuous flow process for the synthesis of a statin side chain precursor by using the DoE approach. Herein, 32.5 °C and pH 7.5 turned out to be the ideal process parameters for DERA (C47M). A series of substrates was tested for its applicability as substrate, but only acetaldehyde **1** and chloroacetaldehyde **2** gave reasonable results. For immobilization alginate was chosen and tested as both alginate beads and alginate-luffa matrix (ALM), of which the latter of them showed a fourfold higher reaction rate, most likely due to an increased surface area. After identifying ALM as a "green" technique for immobilizing biocatalysts, the enzyme was applied in continuous flow. While ALM cross-linked by Ca$^{2+}$ suffered from enzyme leaching, Ba$^{2+}$ led to the sufficiently strong enclosing of the enzyme into the matrix. The usage of freeze-dried cells benefits from the size of the biocatalyst because it can be enclosed into the network more sufficiently. ALM has the major advantage that it can be used for almost all biocatalysts to immobilize them in non-covalent encapsulation. The application is only restricted by the limited stability of alginate against harsh chemicals. However, these chemicals are in most cases not in the application field of biocatalysis. Finally, the optimized flow process (0.1 mL/min, 0.25 M of chloroacetaldehyde, and 0.5 M of acetaldehyde) produced 4.5 g of product per day in a bench-top reactor not bigger than a sheet of paper in area and 15 cm in height. The heterogeneous biocatalyst performed stable for 4 h and convinced by its simple,

inexpensive, and fast preparation. The mutant proved stability over the whole course of the continuous process and in the batch processes (homogeneous and heterogeneous). In addition, the whole catalytic system is biodegradable and made from renewable resources. In order to further increase the yield of the process, a longer reaction time could help. An alternative immobilization method, which is not limited to a certain enzyme loading due to rapidly increasing viscosity, can also be a solution.

**Supplementary Materials:** The following information available online at http://www.mdpi.com/2073-4344/10/1/137/s1, **1.** Figure S1: SDS-PAGE of harvested *E. coli* cells overexpressing DERA; **2.** List of experiments (DoE) for optimization of pH and temperature in batch; **3.** Figure S2: GC-FID spectrum of aldol addition with acetaldehyde-substrates and products (**1**, **1a**, **1b**, and **1c**); Figure S3: GC-FID spectrum of aldol addition with chloroacetaldehyde-substrates and products ((**1**, **2**, **2a**, **2b**, and **2c**); **4.** GC-FID spectra of DoE runs, result of DoE without surface-product formation rate over pH and temperature and response surface view from top; **5.** Coating procedure; **6.** Determination of the residence time distribution in the flow reactor incl. scheme of setup; **7.** Product synthesis in semi-batch for reference; **8.** Product isolation procedure; **9.** NMR of intermediate **2a** and product **2c**; **10.** List of experiments (DoE) for optimization of flow process; **11.** Details on data evaluation in MODDE® for optimization of flow process.

**Author Contributions:** Conceptualization, B.G.; Investigation, B.G. and Y.P.; Methodology, Y.P.; Resources, H.G.-W.; Supervision H.G.-W.; Writing—original draft, B.G.; Writing—review & editing, B.G.; and H.G.-W. All authors have read and agreed to the published version of the manuscript.

**Funding:** This research received no external funding.

**Acknowledgments:** The authors B.G. and Y.P. would like to thank Anna K. Schweiger from the Institute for Molecular Biotechnology at Graz University of Technology for expressing the enzyme and her support when it came to questions regarding biochemistry. Furthermore, all authors thank the research group of Pietruszka at the Research Center Jülich GmbH for providing the plasmid of their DERA mutant.

**Conflicts of Interest:** The authors declare no conflicts of interest

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
