# Peer review of "DERA in Flow: Synthesis of a Statin Side Chain Precursor in Continuous Flow Employing Deoxyribose-5-Phosphate Aldolase Immobilized in Alginate-Luffa Matrix"

_catalysts, doi:10.3390/catal10010137_

Round 1

Reviewer 1 Report

Submitted article is interesting manuscript prepared properly. However, some improvements would increase the value of the study:

1) The statistical analysis of the results should be described in more details in Materials and methods section. Moreover, the error bars could improve the quality of the graphs. 

2) The novelty could be more underlined in the introduction.

3) The latin names of the micro- and macroorganisms should be written in italic. 

4) There is no information about the analytical standards used in GC analysis. Where were they purchased from?

Author Response

1) The statistical analysis of the results should be described in more details in Materials and methods section. Moreover, the error bars could improve the quality of the graphs.

Response: Information was added to section 3.3 (optimization in batch). Details on the optimization in flow are available in ESI. Error bars were added to Figure 7, Figure 8, Figure 9 and Figure 13.

2) The novelty could be more underlined in the introduction.

Response: A brief section was added to the introduction.

3) The latin names of the micro- and macroorganisms should be written in italic.

Response: Thank you for the comment. The font of micro- and macroorganisms were changed.

4) There is no information about the analytical standards used in GC analysis. Where were they purchased from?

Response: All chemicals and analytical standard were purchased from Sigma Aldrich and used as received.  Information added to Section 3.

Reviewer 2 Report

Review catalysts-683374 for the Authors

In my opinion  this work is worth to be published. Reviewed work concerns a very important subject, namely new proposely of synthesis of a statin side chain precursor employing deoxyribose-5-phosphate aldolase. However, the use of barium chloride used to induce cross-linking of alginate  seems to be controversial.  Barium chloride is an extremely toxic substance (H301). It finds application in the chemical industry. As an active agent in the production of pesticides, herbicides, fungicides and insecticides. Hence, the use of this compound in the drug production process raises my concerns. Please, comment on this matter. It is a pity that supplementary materials they were not attached.

Author Response

However, the use of barium chloride used to induce cross-linking of alginate seems to be controversial. Barium chloride is an extremely toxic substance (H301). It finds application in the chemical industry. As an active agent in the production of pesticides, herbicides, fungicides and insecticides. Hence, the use of this compound in the drug production process raises my concerns. Please, comment on this matter. It is a pity that supplementary materials they were not attached.

Response: Thank you for the comment! We are aware of the toxicity of barium. However, barium gave the best results of all the tested bivalent ions in terms of cross-linking and prevention of enzyme leaching. We admit that investigations on the amount of barium found in the product stream are necessary prior to further developing the process for industrial application. Furthermore, the amount of barium in the process set-up is very low (mg range).

Reviewer 3 Report

The proposed manuscript reports continuous flow synthesis of statins by employing immobilized deoxyribose-5-phosphate aldolase (DERA). The complete system is reported for efficient and continuous synthesis of statins. The proposed manuscript is well written. Taking into account the quality and scope of the journal, I would recommend the acceptance of this manuscript to publish in ‘Catalysts’.

Herewith I have added a few comments that will help authors to improve the current manuscript:   

The authors should discuss the reusability of immobilized enzymes. Does alginate-loofa matrix (ALM) immobilized system affect enzyme catalytic activity? Does ALM can be applied for immobilization of other industrially important enzymes? It will help to understand the broad applicability of the proposed system. The authors should discuss the stability of immobilized DERA. In figure 3, the pH value (5.5 to 8.5) on X-axis should be corrected. The authors should add error bars to experimental graphs presented in this manuscript. It will help to understand the repeatability of the proposed system. How the authors avoided the leaching of the enzyme during the continuous flow system reported in this study?

Author Response

The authors should discuss the reusability of immobilized enzymes. Does alginate-loofa matrix (ALM) immobilized system affect enzyme catalytic activity?Does ALM can be applied for immobilization of other industrially important enzymes? It will help to understand the broad applicability of the proposed system. The authors should discuss the stability of immobilized DERA.

Response: Recyclability test in batch were not conduct because already in the first batch it was observed that some alginate was rubbed off the luffa sponge because of stirring. This information was also added to the mauscript.
It is difficult to assess the effect of ALM on the enzyme activity. In order to evaluate that, one would need an similar solid support to sustitute the luffa, what would hardly be possible and an other network would be needed to assess the effect of loofa. This would have exceeded the scope of this work, as the opitimation of the process using experimetal  design was the main intention.

Info about the applicability of ALM for other processes and DERA stability was added to the Conclusion.

In figure 3, the pH value (5.5 to 8.5) on X-axis should be corrected.

Response: Commas were changed to dots.

The authors should add error bars to experimental graphs presented in this manuscript. It will help to understand the repeatability of the proposed system.

Response: Error bars were added to the diagrams in Figure 7, Figure 8, Figure 9 and Figure 13.

How the authors avoided the leaching of the enzyme during the continuous flow system reported in this study?

Response: Interestingly, enzyme leaching was not observed over the time coarse of the continuous flow experiment, at least in case of barium used as cross-linking ion. Calcium seems to led to a looser network. The same behaviour was already observed in previous studies in our lab. We hypophized that whole cells can be immobilized quiet well in the alginate network, due to the size of the catalyst. Furthermore, biocatalyst which might not be bond suffiently was flushed out of the system prior to running the experiments by flushing the backed bed with buffer overnight.